# The Role of Inflammation in Retinal Neurodegeneration and Degenerative Diseases

**DOI:** 10.3390/ijms23010386

**Published:** 2021-12-30

**Authors:** Geetika Kaur, Nikhlesh K. Singh

**Affiliations:** 1Integrative Biosciences Center, Wayne State University, Detroit, MI 48202, USA; hj1044@wayne.edu; 2Department of Ophthalmology, Visual and Anatomical Sciences, Wayne State University School of Medicine, Detroit, MI 48202, USA

**Keywords:** retina, retinal degeneration, AMD

## Abstract

Retinal neurodegeneration is predominantly reported as the apoptosis or impaired function of the photoreceptors. Retinal degeneration is a major causative factor of irreversible vision loss leading to blindness. In recent years, retinal degenerative diseases have been investigated and many genes and genetic defects have been elucidated by many of the causative factors. An enormous amount of research has been performed to determine the pathogenesis of retinal degenerative conditions and to formulate the treatment modalities that are the critical requirements in this current scenario. Encouraging results have been obtained using gene therapy. We provide a narrative review of the various studies performed to date on the role of inflammation in human retinal degenerative diseases such as age-related macular degeneration, inherited retinal dystrophies, retinitis pigmentosa, Stargardt macular dystrophy, and Leber congenital amaurosis. In addition, we have highlighted the pivotal role of various inflammatory mechanisms in the progress of retinal degeneration. This review also offers an assessment of various therapeutic approaches, including gene-therapies and stem-cell-based therapies, for degenerative retinal diseases.

## 1. Introduction

Vision is of critical importance to an individual’s growth and survival. Visual disability negatively impacts productivity, as it reduces independence and mobility [1,2]. Inflammation is the protective response of the immune system to a harmful stimulus and this stimulus could be in the form of toxic metabolites/chemicals, pathogens, damaged cells, physical, traumatic, ischemic, or other challenges [3]. Exposure of these stimuli to pathogen-associated molecular patterns (PAMPs) such as toll-like receptors (TLRs), NOD-like receptors (NLRs), RIG-I-like receptors (RLRs), and damage-associated molecular patterns (DAMPs) upregulates the expression of pro-inflammatory genes and downregulates anti-inflammatory gene expression [4]. Inflammation comprises a cascade of molecular repercussions and cellular activity, which are framed to reinforce a tissue to transform a simple wound or a complex surgical wound, or even to cure severe burn injuries. An unresolved inflammatory cascade will attain an incurable state, which worsens gradually over time, progressing to organ deterioration and diseases [5].

Degenerative retinal diseases are reported as heterogeneous and multiple etiological groups of disorders that hamper the vision of human beings, resulting in compromised quality of life [6]. Retinitis pigmentosa (RP) [7], age-related macular degeneration (AMD) [8], diabetic retinopathy (DR) [9], Stargardt macular dystrophy (STGD) [10], and Leber congenital amaurosis (LCA) [11] are some examples of degenerative retinal diseases. Although the etiology of the mentioned diseases varies, the hallmark characteristic feature of these degenerative retinal diseases is the progressive death of highly differentiated cells within the neurosensory retina, called the retinal pigment epithelium (RPE), or in the photoreceptor cells in the human eye [2,12]. The impairment of the integrity of the retina slowly progresses and finally disintegrates the lining of the retinal cells or causes the death of the specialized light-sensing retinal cells known as photoreceptors that provide visual input to the brain [13]. This cycle worsens over time, resulting in visual deterioration and subsequently blindness or complete vision loss [14].

Inflammatory events are the most likely causes of progressive retinal degenerative conditions [15]. In retinal degenerative diseases, microglia/macrophage are activated, and the activation of these cells leads to the release of cytokines and chemokines for the progressive apoptosis of the photoreceptors suffering from these conditions [16]. Various studies have shown the critical role of macrophages in retinal disease progression. Macrophages change the profile of chemokines and cytokines after accumulation in subretinal space and cause photoreceptor cell death [17]. In this inflammatory cascade, a pivotal role is played by the interleukin-1 (IL-1) family, which comprises of both pro-inflammatory and anti-inflammatory components, especially in retinal degenerative conditions. The proposed roles of IL-1 family members on retinal degeneration are presented in Figure 1. IL-33 is released from the nucleus of endothelial cells, Müller cells, fibroblast or epithelial cells and binds with the heterodimer receptor ST2L/IL-1R accessory protein (IL-1RAcp). The IL-33-ST2L signaling via MyD88-NF-κB-MAPK activation induces the expression of inflammatory cytokines and chemokines [18].

In the present review, we discuss all the genetic and non-genetic causative factors that lead to the progression of retinal degeneration. Additionally, we study the role of inflammation in each retinal disease condition and its corresponding therapeutic strategic approaches based on anti-inflammatory functions.

## 2. Factors Contributing to Inflammation in Retinal Degenerative Diseases

Based on the broad nature of the immune response of retinal degeneration, potential genetic and non-genetic factors play a progressive role in the induction and execution phases of the disease.

### 2.1. Genetic Factors

The influence of genetic factors on retinal disorders is well documented [20]. Genetic factors involve not only pro-inflammatory genes but also anti-inflammatory genes with neuroprotective functions. In the cascade of its initiation to the development of the disease phenotype, several pro-inflammatory genes become activated and are promptly counterbalanced by anti-inflammatory responses [21]. The imbalance in these inflammatory responses results in the progression of degenerative retinal diseases. The expression of genetic factors is regulated both at transcriptional and translational levels to ensure the induction immune response to avoid tissue degeneration [22]. Researchers have witnessed a timely shift in the association of genetic factors with apoptosis to necroptosis and/or pyroptosis for degenerating photoreceptors [23]. The caspase3 activation and apoptosis of photoreceptors are observed in RP mice models such as rds, rd1, and rd10 [24,25,26,27]. The increased expression of RIP1K/RIP3K molecules is linked with necroptosis in P23H-1 rhodopsin rats [23]. The high levels of NLRP3 inflammasome components (NLRP3, active caspase 1, and IL-1β) are associated with increased pyroptosis in murine and canine models of RP [23,28,29].

Substantial evidence in the literature suggests the activation of NLRP3 inflammasome in retinal degeneration through a significant augmentation in the inflammasome components, including *NLRP3*, *ASC*, and *CASP1* [30,31,32]. The caspase-1-dependent pro-inflammatory cytokines, such as IL-1β and IL-18 are also activated in retinal degenerative conditions [33]. The pro-cytokines IL-1β and IL-18 are important regulators of the innate immune system that may cause tissue damage and even cell death. Notably, microglia and infiltrating macrophages are considered the source of the inflammasome activation in degenerating visual cells [19]. Increased expression of IL-1β and IL-18 is found in age-related macular degeneration (AMD) [34], diabetic retinopathy [35], retinitis pigmentosa [36], and glaucoma [37]. IL33/ST2 signaling plays a role in various ocular diseases such as dry eye disease, uveitis, vitreoretinal diseases, and allergic eye disorders [38]. Enhanced retinal cell degeneration and retinal detachment were observed in IL33^−/−^ mice [39].

Several damage-associated molecular patterns (DAMPs) activate inflammasome components in microglia during the dysregulated retinal homeostasis that drives disease progression. In addition, the outer retinal layers are invaded by activated IBA1+microglia/macrophages in diseased conditions. Moreover, the dynamic toll-like receptor 4 (TLR4)—a signaling pathway involved in the activation of NLRP3 inflammasome—is also associated with the pathogenesis of degenerative retinal disorders [31]. The increased expression of *MYD88*, *IRAK4*, and *TRAF6* is also observed in retinal degenerative disorders [22].

Inflammation in degenerative retinal disorders has been strongly supported by molecular genetic studies. Although inconclusive, genes encoding complement factor H (CFH) [40], complement component 2 (C2), factor B (FB) [41], and apolipoprotein E (APOE) have been documented as being associated with AMD. Increased risk of AMD has been especially prevalent among the carriers of the APOE ε2 allele, while APOE ε4 has been known to protect against this condition [42]. Moreover, C2/CFB genes, C3, C9, CFH, and CFI variants in the genes of the complement system are responsible for the progression of AMD [43,44,45].

Epigenetic factors also play a prominent role in the regulation of pro-inflammatory gene promoters (NF-κB and AP-1) involved in retinal diseases. The genes encoding for histone deacetylases (HDACs) and histone acetyltransferases (HATs) are associated with the increased influx of activated microglia in the site of tissue damage, and thus, create a chronic inflammatory cycle, the hallmark of the mentioned disorders [46,47]. Further, microRNA is known to play a biased and remarkable role in driving the M1 phenotype in the mixed microglia/macrophage population in retinal diseases, for example, upregulation of miR-155 [48,49]. Future studies are needed to identify key genes and signaling pathways to understand the pathogenesis of retinal disorders.

### 2.2. Non-Genetic Factors

The heterogeneous and/or genetically complex retinal diseases are triggered by an array of environmental risk factors. Among various environmental factors, age [50], sunlight exposure, smoking, body mass index (BMI), diabetes, alcohol consumption, and other lifestyle-related factors such as physical activity are associated with retinal disorders [51,52]. Epidemiological studies have reported that the prevalence of retinal disorders was 52.37% in subjects aged 60 years and above. Besides age, most of the nongenetic factors are modifiable. Lifestyle and behavioral habits play a significant role in the development of disease [53]. For instance, regular physical exercise has a protective role in various diseases. In the Beaver Dam Eye Study, diabetes was found to be associated with refractive errors [54], and diabetes can be controlled by a healthy diet and exercise. In several other cohort studies, it was observed that constant physical activity may be considered effective for AMD prevention [55].

Further, smoking has been documented to reduce macular pigment concentration by approximately 50% [56] due to the formation of arachidonic acid, which is a precursor of inflammatory mediators like prostaglandins and leukotrienes [57]. The high concentration of hydroquinone known to be present in cigarette tar has also been documented to cause lesions in the eye in murine models [58]. Education, ethnicity, hypertension, hyperthyroidism, Alzheimer’s, and Parkinson’s disease are the other non-genetic factors responsible for retinal diseases, especially AMD [53]. Oxidative stress is a problem during aging and diabetes. Oxidative stress is responsible for the accumulation of ROS that induces lipid peroxidation and glycoxidation, which increases the levels of advanced glycation end products (AGEs) along with advanced lipoxidation end products (ALEs) [59,60]. AGEs and ALEs play a critical role in the chronic inflammatory process and cause alteration in cell signaling, which further causes cell damage and death via NF-κB and MAPK signaling pathways [59,60].

Little is known about the role of non-genetic factors in retinal disorders, as these retinal problems may remain asymptomatic until their advanced stages. Thus, it is difficult to explain the underlying non-genetic risk factor of degenerative retinal disorders, however it can be considered as a correlating risk factor.

## 3. Role of Inflammation in Age-Related Macular Degeneration

Age-related macular degeneration is one of the most studied retinal degenerative disorders. Abbreviated as AMD, it is defined as a slow and steady progressive chronic death of cells such as retinal pigment epithelium (RPE), photoreceptors, Bruch’s membrane, and the choroidal neovascularization in the macula, leading to drusen formation, hypo-, and/or hyperpigmentation [61]. The etiology of AMD remained unclear for more than a century. The identified risk factors include age, smoking of tobacco, fatty food intake, irregular diet, obesity, reactive oxygen intermediates, ethnicity, and heredity [53].

AMD is categorized into two groups based on the time of onset. In the case of early AMD, observable symptoms are inconspicuous and are characterized by the presence of drusen formations at the sub-retinal pigment epithelium [62]. Cases of late AMD are associated with severe loss of vision and have traditionally been classified into “wet” and “dry” forms [62,63]. Among these two forms, “dry” AMD is most common and is characterized by a drusen appearance, i.e., the accumulation of waste products in the retina, that may grow. This may stop the flow of nutrients and thereby cause the death of retinal cells in the macula, causing blurred vision. On the other hand, “wet” AMD, also known as neovascular AMD, is a rapid process of serious vision loss that occurs due to the growth of tiny blood vessels in the retina, which often break or leak. The end-stage of dry AMD, also called geographic atrophy, occurs less frequently than neovascular AMD. This causes degeneration of the macula, due to which the RPE no longer supports the functions of photoreceptors [64].

In most cases of AMD, the larger the area covered by the drusen in early AMD, the greater the chance of developing late-stage AMD [65]. Drusen often remains undetected in early AMD owing to the lesser amount of the area covered initially, as drusen having a diameter of fewer than 25 μm cannot be detected under standard ophthalmoscopy [66].

The innate immune system mediated by mononuclear phagocytes is a major factor in the development of advanced AMD [67]. Retinal microglial cells have been theorized to play a major role in maintaining normal retinal physiology [68,69]. AMD is prominently characterized by the accumulation of microglial cells within the subretinal space [70,71], with greater concentrations around reticular pseudo drusen, a common lesion associated with AMD. This accumulation leads to a range of negative effects on the retinal pigment epithelium and the photoreceptors. The migration of microglial cells and other mononuclear phagocytes from the peripheral circulation plays a major role in retinal degeneration [72]. However, the mechanism of their migration remains under investigation. The release of chemotaxis-mediating chemokines, such as chemokine ligand 2 (Ccl2) has been put forward as a major component in this pathway [73,74], although it is unlikely that this is the sole factor behind microglial migration. CFH is another factor that plays a role in the greater turnover of microglial cells in the subretinal space [75]. Chemokines bind to the CX3CR1chemokine receptors present on the surfaces of the inflammatory cells, including macrophages, microglia, T-cells, and astrocytes. CX3CR1 facilitates the recruitment of WBCs into the inflamed tissues in the retina and thus the inflammatory cells are subsequently activated [76].

The toll-like receptor (TLR4) upregulates interleukin-1β (IL-1β), tissue necrosis factor, and interleukin-6 (IL-6), through the nuclear factor kappa beta (NF-κB) pathway [77,78]. Reports have shown that D299G *TLR4* is a variant that leads to a decline in the elimination of microbial organisms, and low-grade inflammatory changes are behind the pathological changes observed in AMD [79].

An IL8 -251A/T polymorphism has been previously reported in many inflammatory diseases and cancers. A similar association was found between AMD and the homozygous IL8 –251AA genotype [80]. Cytokines, such as IL-6, TNFα, and IL-8, and CRP are responsible for the progression of AMD [81,82]. Seddon et al. [83] reported the correlation between CRP, IL-6, and the disease progression to advanced AMD. The authors observed only a 17% progression rate of AMD in subjects with CRP levels of less than 0.5 mg/L, whereas a significant increase to 38–40% was found in subjects with a CRP range of 0.5–9.9 mg/L. A significantly higher AMD progression rate of 58% was associated with CRP levels greater than 10 mg/L, signifying that CRP levels directly correspond with the AMD progression. Furthermore, the authors also correlated IL-6 levels with AMD progression. They observed no significant changes in the progression rate of AMD with an IL-6 range of <2–5.9 pg/mL, however they did note a significantly increased risk for progression of AMD was found to be associated with IL-6 levels of 6.0 pg/mL or higher. Levels of IL-1α, IL-1β, IL-4, IL-5, IL-10, IL-13, and IL-17 were seen to be markedly elevated in the blood serum samples of subjects who had been diagnosed with advanced stage AMD when compared with those of healthy volunteers [84]. Pro-inflammatory cytokine IL-33 plays a role in both innate and adaptive immune response by activating inflammatory signaling pathways including NF-κB and MAPK signaling to induce the production of pro-inflammatory (such as IL-1β, TNFα, IL-4, IL-6, and CCL2) or anti-inflammatory (like IL-10) cytokines. Further studies proved that RP epithelium cells induce IL-33 signaling and cellular recruitment of microglia and macrophages are controlled by Müller cells into the retina, leading to the destruction of photoreceptors and RPE [19,85].

## 4. Role of Inflammation in Inherited Retinal Dystrophies

Inherited retinal dystrophies (IRD) is an umbrella term used for a host of retinal diseases associated with photoreceptor dysfunction and loss, leading to progressive loss of vision. The most common forms of IRDs include retinitis pigmentosa (RP), Leber congenital amaurosis, Stargardt macular dystrophy, macular degeneration, choroideremia, and Usher’s syndrome [23]. There is much difficulty when attempting to identify the underlying genetic mechanism behind the phenotypes of inherited retinal degeneration, as they are generated through various pathways. The P2X7R upregulation has been shown to enhance inflammasome activation, which leads to the release of proinflammatory cytokines and retinal degenerative diseases [86]. Mutation in Glyoxalase 1 (GLO1) leads to the accumulation of advanced glycation end products (AGE) and retinal degeneration [86]. The accumulation of misfolded proteins increases reactive oxygen species (ROS) generation, enhancing unfolded protein response (UPR) pathways, such as PERK (PKR-like endoplasmic reticulum kinase) and IRE1 (inositol-requiring enzyme 1) pathways in photoreceptor cells resulting in retinal degeneration [86]. An increase in CERKL mutation leads to increased apoptosis and retinitis pigmentosa [86]. Oxidative stress in retinal pigmental epithelial cells alters the expression of micro-RNAs (miRNAs) and long non-coding RNAs (lncRNAs), which induce biochemical pathways involved in RP pathogenesis [86]. Excessive activation of MUTYH leads to the formation of single strand breaks of DNA, causing disturbed homeostasis and cell death [86]. More than 300 genes have been identified so far that can lead to any of the IRDs mentioned above [23]. IRD is a genetic disease and presents high heterogeneity, which results in hard to find a specific mutation. The most frequent of these IRDs is retinitis pigmentosa (RP) with a prevalence of 1 in 3500 individuals [87]. Based on the photoreceptors affected, IRDs can be classified into rod- and cone-dominated dystrophies, and dystrophies encompassing both rods and cones [88]. In the case of rod-dominated dystrophies like RP, night-blindness and gradual loss of peripheral vision are observed, which eventually lead to tunnel vision. Cone-dominated macular dystrophies are characterized by central vision impairment, loss of details, abnormality in color vision, and delay in light to dark adaptation. In IRDs that affect both rods and cones, there is concurrent loss of central and peripheral vision [54].

The involvement of inflammation across different types of IRDs is difficult to generalize, as there is a wide range of inflammatory responses involved in each condition. The role of inflammation in the case of RP, Stargardt macular dystrophy, and Leber congenital amaurosis is discussed in detail in the following sections. The impact of inflammatory responses in the case of IRDs is often due to chronic excessive reaction of the cells and cellular products involved, leading to cell degeneration and apoptosis in the retina [89,90,91,92,93]. Activated microglial cells and macrophages of the immune system secrete cytokines, chemokines, and pro-inflammatory mediators such as TNFα in response to harmful stimuli, tissue disruption, or the presence of free radicals. TNFα regulates various signaling pathways of cell death and survival, which are represented in Figure 2 [23]. The most common cell-signaling pathways that are activated under such circumstances are Janus kinase/signal transducer and activator transcription (JAK-STAT), NF-κB, and the mitogen-activated protein kinase (MAPK) pathways, which then lead to the generation and release of pro-inflammatory interleukins such as IL-1β, IL-6, IL-8, and IL-12 [94,95,96].

## 5. Role of Inflammation in Retinitis Pigmentosa

Retinitis pigmentosa (RP) is an inherited retinal degenerative disease and it often leads to vision loss, as no effective treatment strategies are available [97]. RP is the most common inherited disease of the retina, and it affects 1 in 4000 individuals globally [98]. Research studies indicate that around 69 genes undergo genetic mutations in RP disease that are normally involved in the functioning and maintenance of photoreceptor rod cells [https://sph.uth.edu/retnet/sum-dis.htm (accessed on 30 November 2021)]. Most genes responsible for Retinitis pigmentosa are the same. However, for individual genes, there is a slight difference in the proportion of mutation among different ethnicities. In the Chinese cohort, CYP4V2, RHO, USH2A, RPGR, CRB1, RP2, and CHM are the top seven genes in which a proportion of two out of three mutations occur [99].

There are also some rare RP cases owing to mutations in mitochondrial DNA [100] or due to digenic and diallelic inheritance of *RDS* and *ROM1* genes [101]. Uniparental isodisomy (two identical chromosomes generated from parental homolog in any individual) and incomplete penetrance have also been reported in RP cases [102]. In 1990, *Rhodopsin* was identified as the first gene involved in RP [103]. In addition, RP is caused by gene mutations expressed in the photoreceptor supporting tissue, such as retinal pigment epithelium (RPE). Rod cell apoptosis is followed by ROS generation, which leads to rod and cone cell degeneration in RP [104]. Oxidative stress also affects the expression of microRNA GLO1, CERK-L, MUTHYH, and P2X7R, which cause the release of pro-inflammatory cytokines and ROS production in RP [86,105,106,107]. The molecular mechanisms of oxidative stress in RP are presented in Figure 3.

An oxidative imbalance is found in the eyes of RP patients, which is consistent with findings in various animal models [108,109,110]. Cytokines and chemokines regulate innate and adaptive immune responses, and elevated levels of various pro-inflammatory interleukins (IL1β, IL-2, IL-4, IL-6, IL-8), interferon (IFN)-γ, monocyte chemotactic protein 1, and anti-inflammatory interleukins such as IL-10 in the vitreous humor of patients with RP suggest that these molecules mediate or regulate the immune response in RP. Moreover, increased levels of MCP1 in cells responsible for the activation of microglia and monocyte recruitment, memory cells, and dendritic cells to the site of injury have been detected in both the aqueous and vitreous humors of RP patients [111].

Okita et al. [112] observed increased serum levels of IL-8 and RANTES in patients with retinitis pigmentosa, and they observed a negative correlation between IL-8 expression and central visual function. Inflammatory cytokines (e.g., TNFα, IL-1β, and IL-17) and environmental stresses (e.g., hypoxia and oxidative stress) promote IL-8 expression [113]. In conclusion, activated peripheral inflammatory responses and increased serum IL-8 levels are responsible for central vision in patients with RP.

## 6. Role of Inflammation in Stargardt Macular Dystrophy (STGD)

Stargardt disease is identified as the most predominant variety of inherited macular dystrophy in children and adults with a rate of occurrence of 1 in 8000–10,000. Autosomal recessive mutations in the ATP-binding cassette transporter gene A4 (ABCA4) results in STGD [114,115]. STGD is presented with a heterogeneous phenotype that comprises of clinical and genetic aspects. Genetic modifiers and a few environmental factors predominantly influence the phenotype of STGD [116,117]. The major contribution of microglia in STGD was characterized by Kohno et al. in an *Abca4*/*Rdh8* double knockout mouse model [118]. The available studies are not sufficient to conclude so decisively, however it is evident that TLR4 signaling activates microglia, which further secrete IL-1 family members and eventually regulate STGD pathogenesis. The involvement of chemokine CCL3 is also shown in the progression of retinal degeneration [119,120,121].

## 7. Role of Inflammation in Leber Congenital Amaurosis

Leber congenital amaurosis (LCA) presents a spectrum of hereditary retinal disorders and is a severe congenital/early-onset retinal dystrophy (EORD). Globally, LCA affects around 1 in 8000 children [122]. LCA is reported based on symptoms of severe visual damage at birth or within a few months of the infant’s life and is often presented with roving eye movements or nystagmus. Additionally, LCA is characterized by poor pupillary light sensory responses and oculodigital signs [123]. LCA associated genes along with their functions are mentioned in Table 1 [124].

Researchers are trying to develop an adeno-associated virus (AAV) vector for retinal degenerative diseases treatment. AAV viruses are DNA viruses and can be recognized by the toll-like receptor 9 (TLR-9) [125]. TLR9 activation in RPE cells by CpG-DNA induces the release of pro-inflammatory chemokine (IL-8), which results in the initiation of an inflammatory cascade, where peroxynitrite (ONOO-) induces the upregulation of transcription factors NF-κB and AP1 [126,127]. Preventing this TLR9-initiated inflammatory cascade is crucial for retinal degeneration and for designing retinal gene therapy [125].

## 8. Therapeutic Approaches to Treat Retinal Inflammation

Retinal degenerative diseases present major etiological factors of untreatable blindness worldwide, and efficacious therapeutic options for these conditions are needed. There are many therapeutic approaches for the treatment of neurodegenerative diseases such as stem cell therapy, gene replacement therapy, retinal prostheses, optogenetics, and neuroprotective approaches. Although various therapies are under clinical trials, the use of retinal prostheses has recently received approval [128]. Retinal prostheses help to restore vision in patients suffering from AMD or IRD by the artificial replacement of degenerative photoreceptor cells [128]. Gene replacement therapy, also known as gene addition or gene augmentation, is a method of treating retinal degenerative diseases (IRD). RPE65-LCA is the best example for gene therapy, where an AAV2 vector was used to transfer a normal and healthy RPE65 gene in the retina of LCA patients [129,130,131]. Optogenetic therapy is another therapeutic approach, where “optogenes” are delivered to the target-specific cell types in the retina to enable the cells to become light-sensitive [132,133]. Besides this, neuroprotective methods are also helpful in the treatment of vision loss by increasing the release of neuroprotective molecules and preventing photoreceptor degeneration. Moreover, the advantage of this method is that it does not depend on any specific mutation [134,135]. Optogenetic and neuroprotective therapies are valuable in the treatment of vision loss in RP patients, wherein 50% of the mutations remain unknown [128]. Based on the causality of the disease, AMD treatment strategies can be described through the schematic diagram depicted in Figure 4 [136]. The anti-inflammatory and cell-based therapies are discussed in detail in the present manuscript.

### 8.1. Anti-Inflammatory Therapies

Microglia become activated during neuroinflammation, which is a common feature of retinal degenerative diseases, and produce a myriad of inflammatory cytokines or chemokines in response to stimuli [137]. The treatment of retinal diseases with anti-inflammatory agents is a good approach. Various anti-inflammatory agents used in retina degeneration treatment are listed in Table 2 [136,138,139,140,141,142,143]. Curcumin, or quercetin, is a natural compound with antioxidant and anti-inflammatory properties that has shown positive effects against retinal cell injury [144].

The trans-membrane protein sigma 1 receptor (Sig1R) is a novel target for retinal diseases [145]. Sig1R was initially reported as an opioid receptor, and multiple studies have been conducted in isolated retinal cells, such as microglia, Müller glial cells, astrocytes, and retinal ganglionic cells, and the intact retina [146]. Researchers have provided in vivo evidence of the potent neuroprotective effects of Sig1R against loss of ganglion cells as well as loss of photoreceptor cells [146]. There is evidence from the studies performed over the past two decades that the Sig1R molecule plays a pivotal role in patients with neurodegenerative diseases.

IL-1Ra, otherwise called interleukin-1 receptor antagonist, is a well-known anti-inflammatory competitive receptor antagonist [147]. IL-1Ra prevents the activation of IL-1R by inhibiting the binding to its agonists such as IL-1α and IL-1β, preventing inflammatory activities. Furthermore, mature dendritic cell activation has also been suppressed in mouse cultured RPE cells, by IL-1Ra [145]. Various animal studies have shown the anti-inflammatory role of IL-1Ra, and this might be useful for AMD treatment.

Corticosteroids are shown to have anti-inflammatory properties and are effective against inflammation [148]. The drugs presently used in the treatment of retinal degenerative diseases are listed in Table 1. Many studies have shown that Norgestrel, a synthetic form of the progesterone hormone, can be used for the treatment of retinal degeneration. Norgestrel can inhibit apoptosis and inflammation through the regulation of progesterone receptor membrane component 1 [149]. Another target for the treatment of retinal degeneration diseases is the use of endocannabinoid system receptor agonists (CB1 and CB2), which have a neuroprotective role and anti-inflammatory properties in neurogenerative diseases [150].

### 8.2. Cell-Based Therapies

The subretinal space in the retina is immune-privileged and is used for the delivery of cells for treatment. There are two cell-based therapies, one is stem cell-based therapy, and the other is non-stem cell-based therapy. Stem cell-based therapy involves the transfer of new RPE cells to the subretinal space to improve or maintain the health of the damaged light-sensitive cells. The non-stem cell method involves the implantation of the absent or deficient cells, which can produce protective factors [151].

Microglial cells are involved in the maintenance of retinal structure and function. However, microglial activation is acknowledged as a hallmark of inflammation as they are the resident innate immune cells [93]. These activated cells release neurotoxic pro-cytokines, such as TNFα, IL-1β, and IL-6, and secrete matrix metalloproteinase-9 (MMP-9) [152,153]. The activated form of microglial cells can be observed in AMD, RP, or DR. Therapeutically, microglia inhibitory approaches such as minocycline, resveratrol, or tamoxifen can be considered for the suppression of photoreceptor cell death and pro-cytokine release.

Gene modification to increase the body’s capacity to produce antioxidants is another approach to neutralizing oxidative stress. Usui et al. observed that transgenic over-expression of superoxide dismutase 1 (SOD1) and glutathione peroxidase 4 (GPx4), which catalyze O_2_ and H_2_O_2_, delay the degeneration of cone in rd1 mice [154]. The adeno-associated virus (AAV) vector-mediated delivery of NRF2, a transcription factor that boosts antioxidant gene expression upon oxidative stimulation, is effective for cone survival in rd1, rd10, and rhodopsin^−/−^ mice [155]. The efficacy of antioxidants in animal models with distinct genetic mutations suggests that oxidative stress is a common pathological manifestation of the degenerative retina in RP [156,157].

Many viral vectors (such as adenovirus and lentivirus-based) and nonviral gene transfer methods (nanoparticles and liposomes) have been evaluated for in vivo gene therapy approaches. AAV vector-based therapy is considered the most promising approach for gene therapy in the eye due to its efficiency and stability for gene transfer [158,159]. Luxturna (voretigene neparvovec-rzyl) is a successful example of AAV vector-based gene therapy treatment for RPE65-LCA [125].

## 9. Conclusions and Future Directions

Currently, no definite cure has been discovered for retinal degenerative diseases that lead to the persistent deterioration of vision and ultimately blindness. To find a suitable solution to prevent or target these diseases, many clinical trials are being conducted all over the globe.

While many of these treatment modalities target choroidal perfusion, oxidative stress, and RPE degeneration inflammatory pathways, we could not identify many significant positive results available about the role of inflammation or therapeutic achievements against Leber congenital amaurosis. There is quite a long history of treatment modalities for AMD variants. However, transplanted hESC-derived RPE cells show that AREDS formulation helps in curbing the disease-progression risk of AMD drastically, by about 30%. The safety criteria, such as medium-term and long-term safety, and graft survival in the host environment, are to be explored further in detail.

More clinical studies are required to assess the long-term safety and efficiency of these therapeutic approaches, especially those involving the current hESC-or iPSC-based RPE transplantation strategy. A thorough understanding of the trials and their progression from clinical observation to the laboratory is needed. Future studies from the bench-side to the bedside need to be designed and carried out to better understand the etiological mechanism of these diseases, and to engineer suitable therapies for patients.

## Figures and Tables

**Figure 1 ijms-23-00386-f001:**
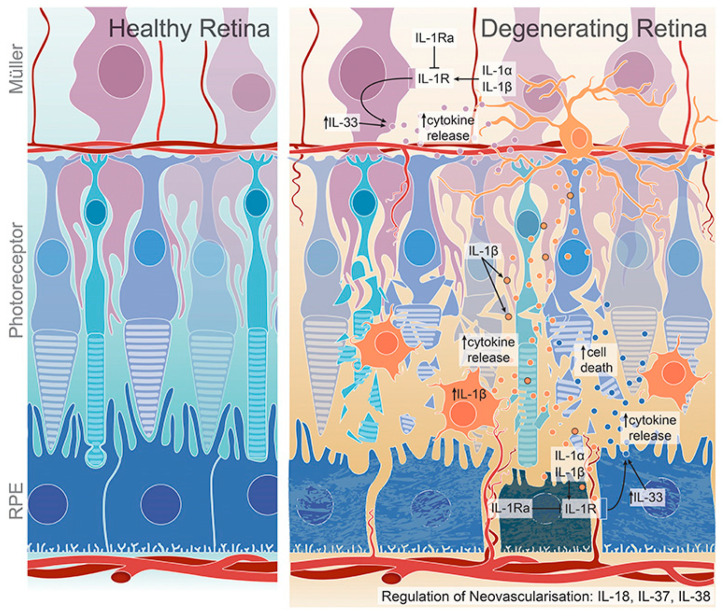
IL-1 family members and their role in retinal degeneration. Activated microglia/macrophages secrete IL-1β, which further induce the release of chemokines and cytokines from Müller and RPE cells, promoting microglia/macrophage recruitment to the inflamed retina. The expression of IL-1Ra, which is a competitive antagonist for IL-1R, is dysregulated in degenerated retina. IL-33 regulates cytokine expression in dry AMD. IL-18, IL-37 and IL-38 have both pro- and anti-angiogenic effects and regulate retinal neovascularization. Adapted from Wooff et al. [19].

**Figure 2 ijms-23-00386-f002:**
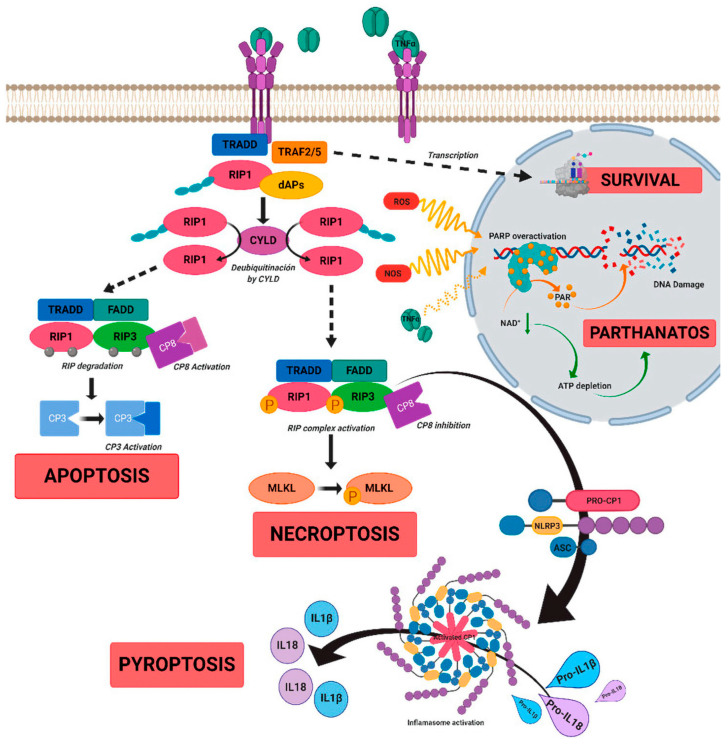
A schematic diagram to show the role of tumor necrosis factor alpha (TNFα)-signaling in inherited retinal dystrophies. TNFα binds to its receptor TNFR1, which leads to the recruitment of death-domain containing adaptor protein (TRADD). TRADD further recruits TNF receptor-associated factor 2 (TRAF2) and receptor-interacting protein kinase 1 (RIPK1) to form complex 1, which is needed for NF-κB activation. Complex 1 dissociates from TNFR1 and associates with Fas-associated protein with death domain (FADD) and pro-caspase 8 to form complex 2. The FADD/caspase 8 association depends on complexes containing unubiquitinated RIPK1 as a scaffold. Activated caspase 8 further induces caspase 3 and apoptosis. TNFα signaling also regulates necroptosis when caspase 8 is not active. RIPK1 recruits RIPK3 to form the necrosome complex. RIPK3 phosphorylates the pseudokinase kinase-like domain of mixed-lineage kinase domain-like (MLKL), leading to its oligomerization. Thus, MLKL recruitment to the plasma membrane induces necroptosis by triggering Ca^+^ and Na^2+^ influx into the cell. RIPK3 also promotes the NLRP3 inflammasome formation and interleukin (IL)-1β activity. TNFα or oxidative stress activates parthanatos through the overactivation of poly [ADP-ribose] polymerase 1 (PARP1). Overactivation of PARP1 leads to decrease in cellular ATP and NAD+ storage, and ultimately to bioenergetic collapse and cell death. Adapted from Olivares-González et al. [23].

**Figure 3 ijms-23-00386-f003:**
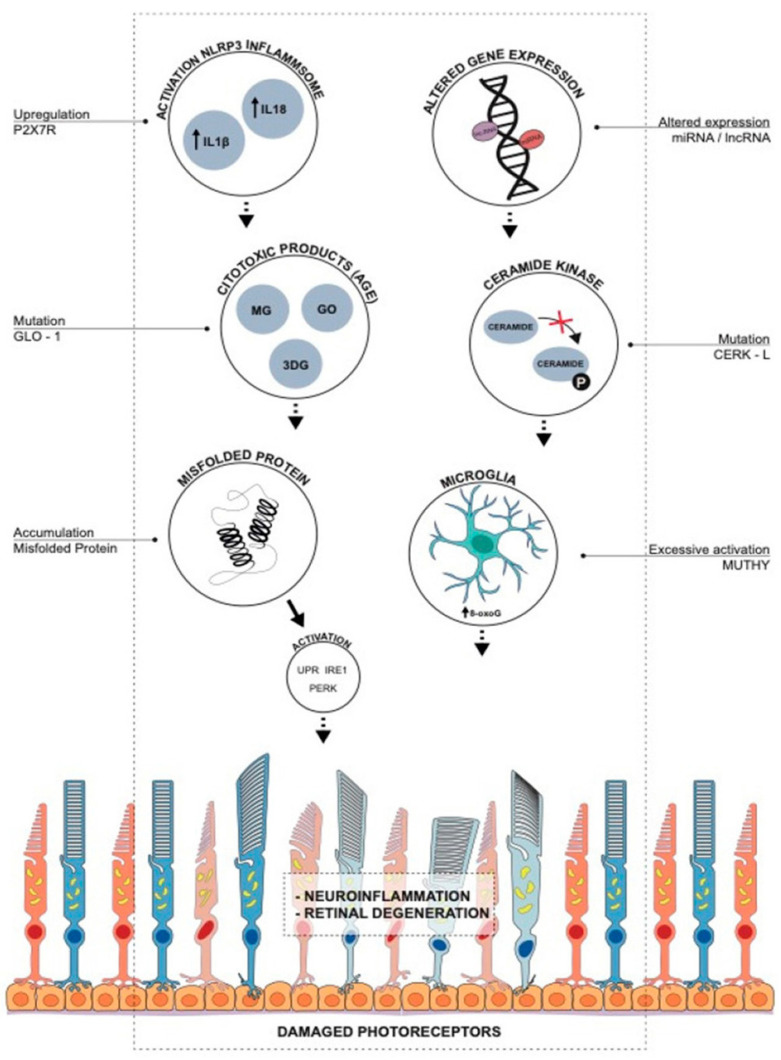
Oxidative stress-induced molecular mechanisms of retinal degeneration in retinitis pigmentosa. Upregulation of P2X7R leads to inflammasome activation, the release of proinflammatory cytokines, and retinal degenerative diseases. Glyoxalase 1 (GLO1) regulates ROS generation and advanced glycation end products (AGE), and mutation in GLO-1 leads to accumulation of AGEs and retinal degeneration. Accumulation of misfolded proteins causes an increase in ROS, enhancing unfolded protein response (UPR), PERK (PKR-like endoplasmic reticulum kinase), and IRE1 (inositol-requiring enzyme 1) pathways in photoreceptor cells leading to retinal degeneration. Ceramide-kinase-like (CERKL) gene induces the phosphorylation of ceramide and protects the cell from oxidative stress-induced apoptosis. An increase in CERKL mutation leads to increased apoptosis and RP. Altered expression of micro-RNAs (miRNAs) and long non-coding RNAs (lncRNAs) by oxidative stress in retinal pigmental epithelial cells induces biochemical pathways involved in RP pathogenesis. MUTYH (mutY DNA glycosylase) is responsible for the maintenance of genomic integrity, and excessive activation of MUTYH leads to the formation of single-strand breaks of DNA, causing disturbed homeostasis and cell death. Adapted from Gallenga et al. [86].

**Figure 4 ijms-23-00386-f004:**
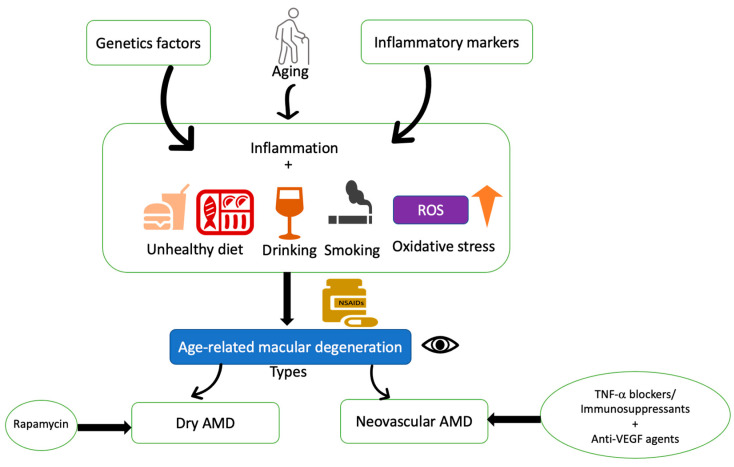
Schematic diagram of therapeutic rationale for non-vascular and neovascular AMD. Age-related macular degeneration is a complex disease affected by various factors, such as genetic factors, aging, inflammatory effects, and oxidative stress. Several risk factors such as unhealthy diet, smoking, alcohol consumption, and obesity are related to oxidative stress. Topical or intravitreal use of non-steroidal anti-inflammatory drugs has shown a preventive effect on AMD. Anti-inflammatory agents/immunosuppressants along with anti-VEGF therapies are currently in use for neovascular AMD treatment. Rapamycin is in clinical trials for Dry AMD.

**Table 1 ijms-23-00386-t001:** Leber congenital amaurosis associated genes and their function.

Gene	Function
GUCY2D	Phototransduction
CRB1 and CRX	photoreceptor morphogenesis
RDH12 and RPE65	retinoid cycle
CEP290	ciliary transport processes

**Table 2 ijms-23-00386-t002:** Currently available treatments for retinal degenerative diseases.

Category	Examples	Properties	Applications	Disease	Reference
Corticosteroids	Dexamethasone Triamcinolone acetonide (TA)	anti-inflammatory, anti-angiogenic, anti-fibrotic, anti-permeability	Used to treat ocular disorders such as macular oedema and angiogenesis	AMD	[136]
Nonsteroidal anti-inflammatory drugs (NSAIDs)	Bromfenac, Nepafenac Diclofenac Aspirin (low dose)	anti-inflammatory, analgesic, antipyretic	Inhibit inflammation Relieve postoperative pain Control countering allergic conjunctivitis and keratitis, inhibit miosis during cataract surgery Reduce cystoid macular oedema	AMD	[138]
Immuno-suppressants	Methotrexate Rapamycin	anti-inflammatory	High doses to treat malignancies Low doses to treat RA without affecting humoral or cellular immunity	AMD	[139,140]
Antibiotics	Tetracyclines	anti-inflammatory properties	Reduce reactive oxygen species Inhibit caspase activation Reduce cell damage and prevent cell death Prevent complement activation, Inhibit matrix metalloproteinases that breakdown the barrier between the RPE and Bruch’s membrane Inhibit cytokine production by regulating microglia and T-cell activation	AMD	[141,142]
Anti-TNFα agents	infliximab, adalimumab, or etanercept	pro-inflammatory	Reduce photoreceptor cell death. Improves the survival of retinal cells in case of glaucoma, and choroidal neovascularization	RP	[29,143]

## Data Availability

Data available in a publicly accessible repository.

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
