# Peer review of "The Role of Inflammation in Retinal Neurodegeneration and Degenerative Diseases"

_ijms, 2021, doi:10.3390/ijms23010386_

Round 1
Reviewer 1 Report
The present work reviews in an appropriate way degenerative retinal diseases, with particular emphasis on inflammatory stimuli on the etiology of these disorders. I found of particular interest the dissertation of the different pathways involved in several retinal disorders and their contribution to the initiation and progression of the pathology.
I only have minor concerns to point out:
-The images styles differ (despite the fact that their provenience is reported) and it might be useful to standardize them in order to present such nice work in a more elegant manner.
- In lines 88-89 remove italics when not needed.
Author Response
Reviewer #1
- The images styles differ (despite the fact that their provenience is reported) and it might be useful to standardize them in order to present such nice work in a more elegant manner.
Answer: In response to the reviewer’s suggestions, we standardized
the figures.
- In lines 88-89 remove italics when not needed.
Answer: Corrected.

Reviewer 2 Report
Reviewer comments and suggestions
In the current review, the authors provided a narrative review of various studies carried out till now on the role of inflammation on human retinal degenerative diseases. Moreover, the manuscript also covers various inflammatory mechanisms in the progress of retinal degeneration. This review encourages researchers who are working in the field of degenerative retinal diseases to understand the modern methods in assessing various therapeutic approaches including gene-therapies and stem-cell-based therapies.
Decision: Minor comments
Below are the comments for this paper to be incorporated in the revised version of the manuscript.
- In abstract, the authors need to mention all the related diseases of RN such as AMD
- Line 72 Why it was italic, (induction and execution) please check
- The genetic factors section, second para. I found some words are italic which was not needed
- Line 137 please check whether a line was not or it should be non-genetic factors because the para was discussing non-genetic factor
- Line 261-263 Please mention a reference for this
- Figure 3 The legend should be described here
- Line 327 The therapy author mentioned here are of two types, is there is any other way, please discuss in the section here
- Line 330 please avoid these kinds of sentences, without references not good to write like this
- Table 1 it would be nice if the drugs were classified based on the explored diseases in the MS
- Line 348-349 Where is the reference for this
- The journal-style used in the MS was not following the journal guidelines. Please check
Author Response
Reviewer #2
- In abstract, the authors need to mention all the related diseases of RN such as AMD.
Answer: Included in the revised manuscript.
- Line 72 Why it was italic, (induction and execution) please check
Answer: Corrected.
- The genetic factors section, second para. I found some words are italic which was not needed
Answer: Corrected.
- Line 137 please check whether a line was not, or it should be non-genetic factors because the para was discussing non-genetic factor
Answer: Corrected in the revised manuscript.
- Line 261-263 Please mention a reference for this
Answer: We have included the reference in the revised manuscript.
- Figure 3 The legend should be described here
Answer: We have included the legend in the revised manuscript.
- Line 327 The therapy author mentioned here are of two types, is there is any other way, please discuss in the section.
Answer: As per the reviewer’s suggestions, we included other therapies in
the revised manuscript.
- Line 330 please avoid these kinds of sentences, without references not good to write like this
Answer: We have removed it from the revised manuscript.
- Table 1 it would be nice if the drugs were classified based on the explored diseases in the MS.
Answer: In response to the reviewer’s suggestions, we have included the
disease section in the revised Table 2 (we have included a new Table 1).
- Line 348-349 Where is the reference for this
Answer: The reference is now included in the revised manuscript.
- The journal style used in the MS was not following the journal guidelines. Please check
Answer: We have revised the manuscript to match the journal style.

Reviewer 3 Report
In this research article, Kaur et.al., reviews an update on retinal neurodegeneration and the role of inflammation on various retinal degenerative diseases. This review is a valuable addition to the field as it summarizes the existing literature on the role of inflammation on retinal degenerative diseases and offers an assessment of various therapeutic approaches. This review has interesting observations which are beneficial to researchers in the areas of retinal neurodegeneration and associated retinal complications. However, the manuscript is quite confusing in parts, with spelling, grammar, syntax of sentences and references needs to be revised in several sections with a thorough proof check of the review. Also avoid repeats of already mentioned abbreviations in the text. Eg. AMD, RPE, DAMPs, RP etc. Some of the other major and minor concerns are discussed below.
Major Corrections:-
Line 2: The title is bit vague and this review will be much better with a title like “The role of inflammation in retinal neurodegeneration and degenerative diseases”
Line 82-84: Explain further and give examples of association of genetic factors with apoptosis to necroptosis and/or pyroptosis for degerating photoreceptors.
Line 94: Also add information and explain about regulation of cytokine expression by IL-33 as its mentioned in the caption of Figure 1. Recent article by Augustine et.al., will be beneficial and good reference here (J Neuroinflammation 16, 251 (2019). https://doi.org/10.1186/s12974-019-1625-y).
Line 137:140: Need to discuss more about non-genetic factors and its possible role in accumulation of Advanced glycation end products (AGEs) and Advanced lipoxidation end products (ALEs) in the retina causing inflammation, along with appropriate references. Also add other possible effects of non-genetic factors in various pathways of inflammation.
Line 203-209: Need to proofread and rephrase these sentences and statements with original articles as references. Also include more information on IL-1a serum levels and IL-33 signalling.
Line 215-217: Need to rephrase and explain further on differences in underlying genetic mechanism and pathways of inherited retinal degenerations.
Line 261 and 318: Add prevalence of RPs and Leber congenital Amaurosis, as similar to STGD section.
Line 309-315: Need to proofread and rephrase these sentences. Also explain further on the role of IL-1b and IL-18 in each of the diseases on Line 314 with appropriate original references.
Line 323-325: Explain further on the inflammatory cascade and progression to blindness with appropriate original references.
Line 391: Mention and describe about the Gene Therapy for Leber Congenital Amaurosis Caused by RPE65 Mutations. Also mention other recent in vivo gene therapy studies for retinal inherited diseases.
Minor Corrections:-
Line 39: Change “the” to “The”.
Line 41: Change “degenerated” to “degenerative”
Line 51: Delete “cells”
Line 98: Clarify the full form of TLR4
Line 203: Rephrase “It is studied”.
Author Response
Reviewer #3
Major Corrections:
- Line 2: The title is bit vague, and this review will be much better with a title like “The role of inflammation in retinal neurodegeneration and degenerative diseases”
Answer: In response to the reviewer’s suggestions, we have now changed
the title of the manuscript.
- Line 82-84: Explain further and give examples of association of genetic factors with apoptosis to necroptosis and/or pyroptosis for degenerating photoreceptors.
Answer: We have now included it in the revised manuscript (Kindly refer to
page 3, lines 183-188)
- Line 94: Also add information and explain regulation of cytokine expression by IL-33 as it's mentioned in the caption of Figure 1. Recent article by Augustine et.al., will be beneficial and good reference here (J Neuroinflammation 16, 251 (2019). https://doi.org/10.1186/s12974-019-1625-y).
Answer: We have added the information in the revised manuscript (Kindly
refer to page 3, lines 198-201)
- Line 137:140: Need to discuss more about non-genetic factors and its possible role in accumulation of Advanced glycation end products (AGEs) and Advanced lipoxidation end products (ALEs) in the retina causing inflammation, along with appropriate references. Also add other possible effects of non-genetic factors in various pathways of inflammation.
Answer: In response to the reviewer’s suggestions, we have now included
it in the revised manuscript (Kindly refer to page 4, lines 401-414)
- Line 203-209: Need to proofread and rephrase these sentences and statements with original articles as references. Also include more information on IL-1a serum levels and IL-33 signaling.
Answer: We have now added the information in the revised manuscript
(Kindly refer to page 5, lines 531,532, 534 & 535; page 6, lines 738 &
739).
- Line 215-217: Need to rephrase and explain further on differences in underlying genetic mechanism and pathways of inherited retinal degenerations.
Answer: We have now added the information in the revised manuscript
(Kindly refer to page 6, lines 750-762).
- Line 261 and 318: Add prevalence of RPs and Leber congenital Amaurosis, as similar to STGD section.
Answer: We have now added the information in the revised manuscript
(Kindly refer to page 7, lines 973 & 974; page 10, lines 1208 & 1209).
- Line 309-315: Need to proofread and rephrase these sentences. Also explain further the role of IL-1b and IL-18 in each of the diseases on Line 314 with appropriate original references.
Answer: We removed these lines from the revised manuscript, as we have
already discussed them in the introduction section.
- Line 323-325: Explain further on the inflammatory cascade and progression to blindness with appropriate original references.
Answer: We have now added the information in the revised manuscript
(Kindly refer to page 10, lines 1217-1223).
- Line 391: Mention and describe about the Gene Therapy for Leber Congenital amaurosis Caused by RPE65 Mutations. Also mention other recent in vivo gene therapy studies for retinal inherited diseases.
Answer: We have now incorporated the information in the revised
manuscript (Kindly refer to page 10, lines 1227-1242; page 13, lines 1628-
1633).
Minor Corrections:
- Line 39: Change “the” to “The”.
Answer: Changed.
- Line 41: Change “degenerated” to “degenerative”
Answer: Changed.
- Line 51: Delete “cells”
Answer: Deleted.
- Line 98: Clarify the full form of TLR4
Answer: Full form is included.
- Line 203: Rephrase “It is studied”.///
Answer: Rephrased.

Round 2
Reviewer 3 Report
Kaur & Singh have addressed most of the concerns raised in the initial manuscript. The revised review reads well, and the efforts are greatly appreciated. This review can deserve publication after the minor changes below:-
Line 117: Update “receptord” to “receptor”.
Line 167: Also add appropriate references for studies of Advanced glycation end products and Advanced lipoxidation end products in the retina.
Line 322: Check and update the website link as it is not accessible.
Figure 4: Update “Age Macular Degeneration” to Age-related macular degeneration in the figure. Also enhance the figure caption by adding more explanation.
Abbreviations: IL-33 and STGD are missing in the list. WBC is repeated. Also, proof check for any other abbreviations which are missing from the list.
Author Response
Reviewer #3
Minor Corrections:
- Line 117: Update “receptord” to “receptor”.
Answer: Corrected.
- Line 167: Also add appropriate references for studies of Advanced glycation end products and Advanced lipoxidation end products in the retina.
Answer: added.
- Line 322: Check and update the website link as it is not accessible.
Answer: updated.
- Figure 4: Update “Age Macular Degeneration” to Age-related macular degeneration in the figure. Also enhance the figure caption by adding more explanation.
Answer: Updated (Kindly see Page 11, 403-408).
- Abbreviations: IL-33 and STGD are missing in the list. WBC is repeated. Also, proof check for any other abbreviations which are missing from the list.
Answer: added.
